# Synergistic and low adverse effect cancer immunotherapy by immunogenic chemotherapy and locally expressed PD-L1 trap

Wantong Song[1,2], Limei Shen[1], Ying Wang[1], Qi Liu[1], Tyler J. Goodwin[1], Jingjing Li[3], Olekasandra Dorosheva[3], Tianzhou Liu[4], Rihe Liu[3,5] & Leaf Huang[1]

Although great success has been obtained in the clinic, the current immune checkpoint inhibitors still face two challenging problems: low response rate and immune-related adverse effects (irAEs). Here we report the combination of immunogenic chemotherapy and locally expressed PD-L1 trap fusion protein for efficacious and safe cancer immunotherapy. We demonstrate that oxaliplatin (OxP) boosts anti-PD-L1 mAb therapy against murine colorectal cancer. By design of a PD-L1 trap and loading its coding plasmid DNA into a lipid-protamine-DNA nanoparticle, PD-L1 trap is produced transiently and locally in the tumor microenvironment, and synergizes with OxP for tumor inhibition. Significantly, unlike the combination of OxP and anti-PD-L1 mAb, the combination of OxP and PD-L1 trap does not induce obvious Th17 cells accumulation in the spleen, indicating better tolerance and lower tendency to irAEs. The reports here may highlight the potential of applying PD-L1 inhibitor, especially locally expressed PD-L1 trap, in cancer therapy following OxP-based chemotherapy.

[1] Division of Pharmacoengineering and Molecular Pharmaceutics, Eshelman School of Pharmacy, University of North Carolina, Chapel Hill, NC 27599, USA. [2] Key Laboratory of Polymer Ecomaterials, Changchun Institute of Applied Chemistry, Chinese Academy of Sciences, Changchun 130022, China. [3] Division of Chemical Biology and Medicinal Chemistry, Eshelman School of Pharmacy, University of North Carolina, Chapel Hill, NC 27599, USA. [4] Department of Gastrointestinal Surgery, The Second Hospital of Jilin University, Changchun 130041, China. [5] Carolina Center for Genome Sciences, University of North Carolina, Chapel Hill, NC 27599, USA. Correspondence and requests for materials should be addressed to R.L. (email: rliu@email.unc.edu) or to L.H. (email: leafh@email.unc.edu)

Checkpoint blockade immunotherapies targeting T-cell co-inhibitory signaling pathways are redefining cancer therapy. Recently, the US FDA has granted accelerated approval to pembrolizumab (Keytruda®, anti-PD-1 antibody) for patients with microsatellite instability (MSI)-high or mismatch repair (MMR)-deficient solid tumors, setting an important first in cancer community for approval of a drug based on a tumor's biomarker without regard to the tumor's original location. However, MSI-high/MMR-deficiency only occurs in a very small fraction of tumor cases. For example, in colorectal cancer patients, only 5% of the population belongs to MSI-high type, while the majority of the population—around 95%—has microsatellite-stable (MSS) or MMR-proficient disease, and does not response to PD-1/PD-L1 based immunotherapy[1].

A major difference between MSI-H and MSS tumor is the lymphocyte infiltration status. MSI is a condition of genetic hypermutability that results from impaired DNA MMR function. Thus MSI-H/MMR-deficient tumors have much more somatic-mutations than MSS/MMR-proficient tumors. The frequency of somatic-mutations within a tumor type is largely correlated with lymphocyte infiltration, as well as sensitivity to immune check-point inhibitors[2]. Therefore, how to improve the antigen-recognition efficiency and lymphocyte infiltration in non-hypermutated MSS/MMR-proficient tumors is a key issue to improve the responses to checkpoint blockade immunotherapies.

Immunogenic cell death (ICD) is a form of cell death caused by some chemo agents such as anthracyclines, oxaliplatin (OxP), and bortezomib, or by radiation and photodynamic therapy[3]. Unlike normal apoptosis, ICD can induce immune responses through activation of dendritic cells (DCs) and consequent activation of specific T-cell responses. This is accompanied by a sequence of changes in the composition of the cell surface, as well as release of soluble mediators, operating on a series of receptors expressed by DCs to stimulate the presentation of tumor antigens to T cells. For example, exposure of calreticulin (CRT) on dying cell surface in ICD promotes the uptake of dead cell-associated antigens, and the release of large amounts of adenosine triphosphate (ATP) and high-mobility group box 1 protein (HMGB1) into the extra-cellular milieu favors the recruitment of DCs and their activation[4]. By this way, ICD promotes antitumor immune responses and increases engulfment of tumor antigens, thus may boost responses of the non-hypermutated MSS/MMR-proficient tumors to PD-1/PD-L1 inhibitor therapy.

To address this hypothesis, we start with an orthotopic col-orectal cancer model, and prove that OxP would boost tumor responses to PD-L1 mAb treatment. In order to reduce the immune-related adverse effects (irAEs) of systematically injected anti-PD-L1 mAb, an engineered PD-L1 trap is designed and its coding plasmid DNA is targeted delivered via lipid-protamine-DNA (LPD) nanoparticles to locally and transiently produce PD-L1 trap fusion protein in the tumor tissue. The combination of OxP and locally expressed PD-L1 trap result in synergistic anti-tumor efficiency with low adverse effects. Similar synergistic antitumor effects are observed in two other non-hypermutated melanoma and breast cancer models. Finally, we analyze color-ectal cancer patient samples and propose that the combination of locally expressed PD-L1 trap and OxP-based chemotherapy may be meaningful for non-hypermutated MSS/MMR-proficient cancer therapy.

## Results

### Establishment of an orthotopic colorectal tumor model.
CT26 cell line was derived from BALB/c mice in 1970s after repeated rectal administration of the carcinogen *N*-nitro-*N*-methylur-ethane[5]. Previous genomic characterization of this cell line showed mutation in *Kras* and lack of mutations in *Apc*, *Tp53*, *Braf*, *Pold1*, and MMR genes *Mlh1*, *Msh2*, *Msh6*, and *Pms2*, indicating CT26 cell line represents a MMR-proficient colorectal cancer type[6,7]. In this work, an orthotopic, syngeneic colorectal cancer model was established by injecting CT26-FL3 cells into the cecum wall of BALB/c mice. CT26-FL3 is a highly metastatic subtype of CT26 cells with high-tumor formation rate and spontaneous liver metastasis potential when implanted orthotopically[8]. The CT26-FL3 cells were stably transfected with firefly luciferase (Luc), therefore, the tumor burden can be monitored by intraperitoneal injection of D-luciferin, followed by bioluminescence analysis using an IVIS system (Fig. 1a). After three injections of anti-PD-L1 mAb, the established CT26-FL3 tumor showed almost no responses to the treatment (Fig. 1b). We performed the same treatment on another orthotopic colorectal cancer model established with MC38 cells, a hypermutated cell line with missense mutations in *Tp53*, *Braf*, *Pold1* and MMR gene *Msh3*[9,10], and a 74.3% tumor sup-pression rate (TSR%) was obtained after the same treatment (Supplementary Fig. 1a, b). Further analysis on these two tumor models showed that there was certain amount of T cells inside the MC38 tumor (Supplementary Fig. 1c), while those inside the CT26-FL3 tumor were minimal (Fig. 1c). These results confirm that MMR gene status and T-cell infiltration are associated with the responsiveness to checkpoint inhibitor therapy in colorectal cancer models, and CT26-FL3 orthotopic tumor appears to be a valid model for MMR-proficient colorectal cancer study.

### OxP induces ICD and immune responses in CT26-FL3 tumor.
To overcome the lack of T-cell infiltration inside the orthotopic CT26-FL3 tumor, we hypothesized that therapeutically induced ICD of the tumor may aid in reversing the tumor's inert immune microenvironment. OxP, a chemo drug used as first-line che-motherapy for colorectal cancer, was reported to induce ICD in various cancer cell lines[11,12]. Using CRT exposure and HMGB1 release as surrogate markers for drug-induced tumor cell immuno-genicity, we found that OxP treatment indeed induced immunogenic effects on CT26-FL3 cells (Supplementary Fig. 2a, b). To identify the host response, BALB/c mice received subcutaneous injection of CT26-FL3 cells pre-incubated with OxP. Seven days later, the spleens were harvested, splenocytes were collected, and an ELISpot analysis was carried out against β-gal and AH1 peptides. AH1 peptide sequence is a H2-Ld-restricted epitope derived from envelope gly-coprotein 70 (gp-70), encoded by the endogenous murine leukemia virus which is universally expressed in CT26 and numerous other murine tumor cell lines[13,14]. Compared to the splenocytes from the control mice (without OxP incubated cells injection), splenocytes from the OxP-treated cells injected group showed obvious IFN-γ production under antigen stimulation, indicating that subcutaneously injected CT26-FL3 cells after OxP treatment induced a systemic immune response against the cell antigen (Supplementary Fig. 2c). Furthermore, the mice receiving OxP-treated cells were re-challenged with $5 \times 10^5$ live CT26-FL3 cells on the opposite flank. Significantly, 2 out of 5 mice showed no tumor appearance, and the remaining 3 mice showed much lower tumor burden compared to the control group (Supplementary Fig. 2d). These results confirmed that ICD occurred when CT26-FL3 cells were treated with OxP. Based on these observations, we administered the orthotopic CT26-FL3 tumor-bearing mice with OxP at a well-tolerated dose (Fig. 2a). OxP treatment significantly increased HMGB1 and CRT staining within the orthotopic CT26-FL3 tumor tissues, a result that mirrors the in vitro findings (Fig. 2b). Furthermore, when the splenocytes from the OxP-treated tumor-bearing mice were stimulated with AH1 peptide, obvious IFN-γ production was observed, while the spleno-cytes from the PBS-treated tumor-bearing mice had no responses (Fig. 2c). These data demonstrate that OxP treatment can effectively

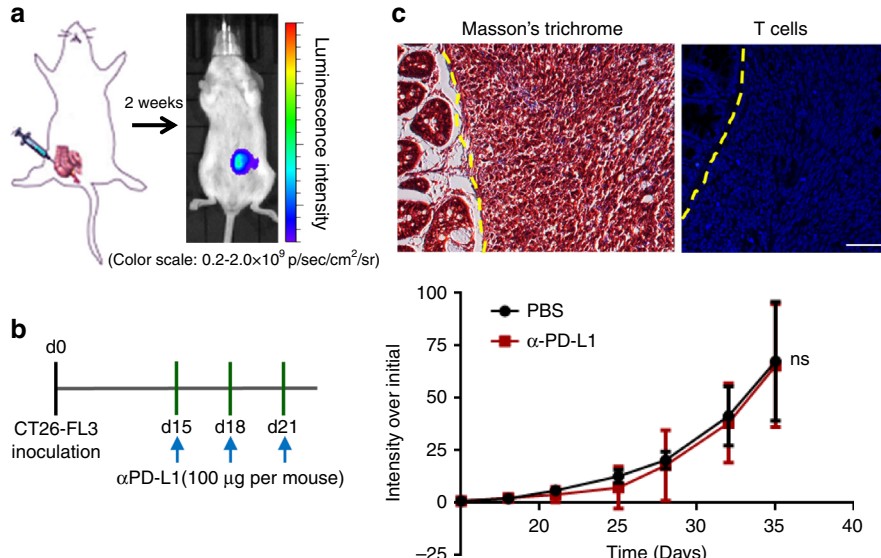

**Fig. 1** Orthotopic CT26-FL3 tumor is resistant to anti-PD-L1 mAb therapy. **a** Establishment of orthotopic colorectal tumor models. CT26-FL3 (RFP/Luc) cells were inoculated into the mouse cecum wall, and the tumor burden was monitored by bioluminescent analysis. **b** Treatment scheme and tumor growth curves of orthotopic CT26-FL3 tumors in PBS and α-PD-L1 treated groups ($n = 5$ mice per group). **c** Masson's trichrome and immunofluorescence staining of orthotopic CT26-FL3 tumor tissues in PBS group using 4′,6-diamidino-2-phenylindole (DAPI, blue) and anti-CD3 antibody (red). Yellow dotted line indicates the border between intestinal mucosa and the orthotopic tumor. Scale bar represents 50 μm. Significant differences were assessed in **b** using two-way ANOVA. Results are presented as mean (SD). ns, not significant

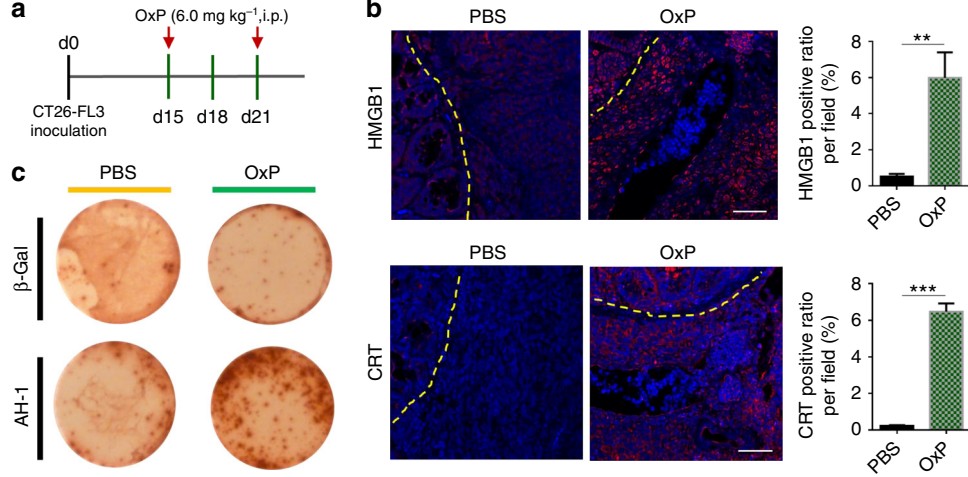

**Fig. 2** OxP induces ICD in the CT26-FL3 tumor. **a** OxP treatment scheme. **b** Immunofluorescence staining of orthotopic CT26-FL3 tumors after PBS and OxP treatment using DAPI (blue), anti-HMGB1 antibody (red), and anti-CRT antibody (red). Positive ratios were quantified in 5 randomly selected fields per mouse ($n = 4$ mice per group). Yellow dotted line indicates the border between intestinal mucosa and the orthotopic tumor. Scale bar represents 50 μm. **c** ELISpot test of the splenocytes of CT26-FL3 tumor-bearing mice after PBS and OxP treatment. Significant differences were assessed in **b** using $t$ test. Results are presented as mean (SD). **$P < 0.01$, ***$P < 0.001$

induce immunogenic phenotypes in CT26-FL3 tumor and significantly promote the antigen-recognition efficiency in vivo.

**Combination of OxP and anti-PD-L1 mAb**. Because of OxP inducing ICD in the orthotopic CT26-FL3 tumor, we wonder the effect of combination of OxP with anti-PD-L1 mAb in CT26-FL3 tumor therapy. We first examined the changes in the tumor immune microenvironment and whether such changes promoted an antitumor immune response. By collecting single-cell suspensions of the orthotopic CT26-FL3 tumors, we found increased amounts of CD8$^+$ T cells, CD4$^+$ T cells, activated DCs and

elevated PD-L1 expression in the tumor of OxP-treated mice as compared to PBS-treated mice (Fig. 3a). In addition, the mRNA levels for various pro-inflammatory cytokines including CCL2, CXCL12, and CXCL13 were significantly increased. CXCL9 and CXCL10, two important cytokines favoring T-cell infiltration into tumor[15], were also obviously increased (Fig. 3b). Th1-type cytokines IFN-γ and TNF-α were significantly elevated after OxP treatment. However, Th2 type cytokines IL-4 and IL-10 (especially IL-10) levels were also greatly increased (Fig. 3b). In further assessing the distribution of immune cells in orthotopic CT26-FL3 tumors by immunofluorescence analysis, we found the appearance of ectopic lymphoid-like structures (ELSs) in the

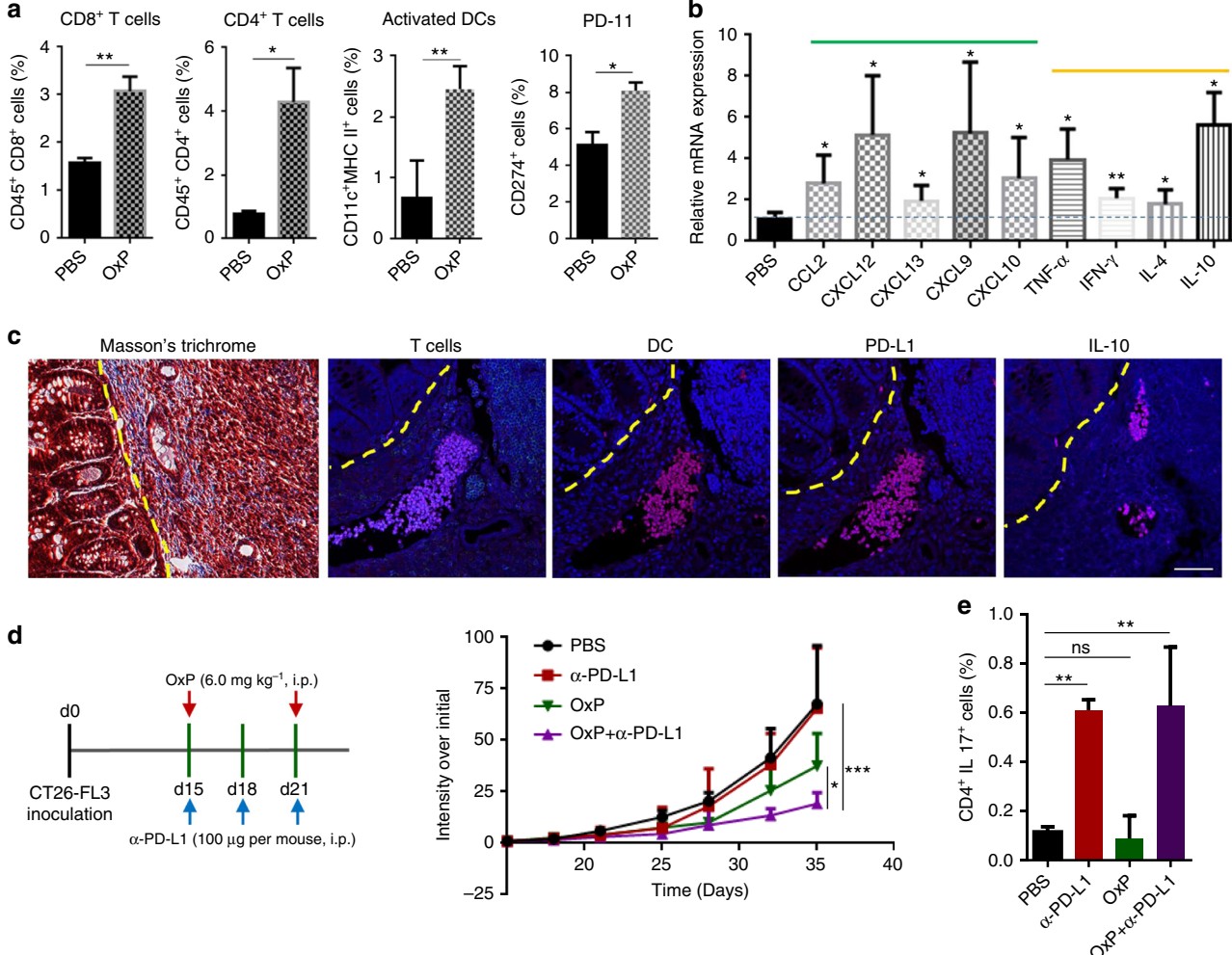

**Fig. 3** OxP induces immune microenvironment changes and synergizes with α-PD-L1 in CT26-FL3 tumor therapy. **a** CD8[+] T cells, CD4[+] T cells, activated DCs and PD-L1 levels in tumors of the PBS and OxP-treated groups on day 28, analyzed by flow cytometry ($n = 4$). **b** Relative mRNA expressions of various cytokines in tumors of the OxP-treated group compared to PBS group on day 28, detected by quantitative RT-PCR ($n = 4$). **c** Masson's trichrome and immunofluorescence staining of the orthotopic tumors after OxP treatment using DAPI (blue), anti-CD3 (red), anti-CD11c (red), anti-CD274 (red), and anti-IL-10 (red). Yellow dotted line indicates the border between intestinal mucosa and the orthotopic tumor. Scale bar represents 50 μm. **d** Treatment scheme and tumor growth curves of orthotopic CT26-FL3 tumors in PBS, α-PD-L1, OxP, and OxP + α-PD-L1 treated groups ($n = 5$ mice per group). **e** Th17 cell ratios in the splenocytes of the mice after various treatments on day 28 ($n = 4$). Significant differences were assessed in **d** using two-way ANOVA with multiple comparisons and in **a**, **b**, and **e** using $t$ test. Results are presented as mean (SD). ns not significant. $*P < 0.05$, $**P < 0.01$, $***P < 0.001$

tumor after OxP treatment. In the ELSs, there were large amount of T cells, DCs, macrophages, natural killer (NK) cells, as well as regulatory T cells (Tregs) and myeloid-derived suppressor cells (MDSCs). In addition, these ELSs were positive with PD-L1 and IL-10 (Fig. 3c, Supplementary Fig. 3). ELSs have been observed in many cancers, and the occurrence of ELS predicts good prognosis in clinical colorectal cancer[16–18]. However, the abundant checkpoint proteins and immunosuppressive cytokines may impair the biological functions of T cells inside these ELSs. We then treated the CT26-FL3 tumor-bearing mice with combination of OxP and anti-PD-L1 mAb. Significant tumor inhibition result was observed in the combination group compared to the other treatments (Fig. 3d, Supplementary Fig. 4), suggesting the synergistic effect of combining ICD and checkpoint inhibitor in MSS/MMR-proficient colorectal cancer therapy. It has been reported that IL-17 and Th17 cells are highly up-regulated in inflammatory tissues of autoimmune diseases, suggesting the ratio of Th17 cells can be used as a parameter to monitor the irAEs of checkpoint inhibitor immunotherapy[19,20]. We examined the Th17 cell ratios in the spleens after various treatments.

Certain amount of Th17 cells were observed in the spleen of anti-PD-L1 mAb and OxP+anti-PD-L1 mAb treatment groups (Fig. 3e), suggesting the possibility of irAEs when applying systematically injected anti-PD-L1 mAb in cancer therapy.

**Design of a locally and transiently expressed PD-L1 trap**. The systemic blockade of immune co-inhibitory signaling pathways could result in severe side toxicities. The adverse effects such as autoimmune disorder/inflammation have been observed in the clinical application of anti-PD-1/PD-L1 mAbs[21–23]. Indeed, increased accumulation of Th17 cells in the spleen was observed in our treatments using anti-PD-L1 mAb. We address this challenge by locally blocking the PD-L1 signaling using an engineered trap fusion protein that potently and specifically binds to PD-L1 and disrupts its biological functions. Specifically, we developed a unique trimeric PD-L1 trap protein by genetically fusing the extracellular domain of PD-1 with a robust trimerization domain from cartilage matrix protein through an optimized hinge linker (Fig. 4a). The trimeric trap, which is efficiently formed from the monomeric trap through

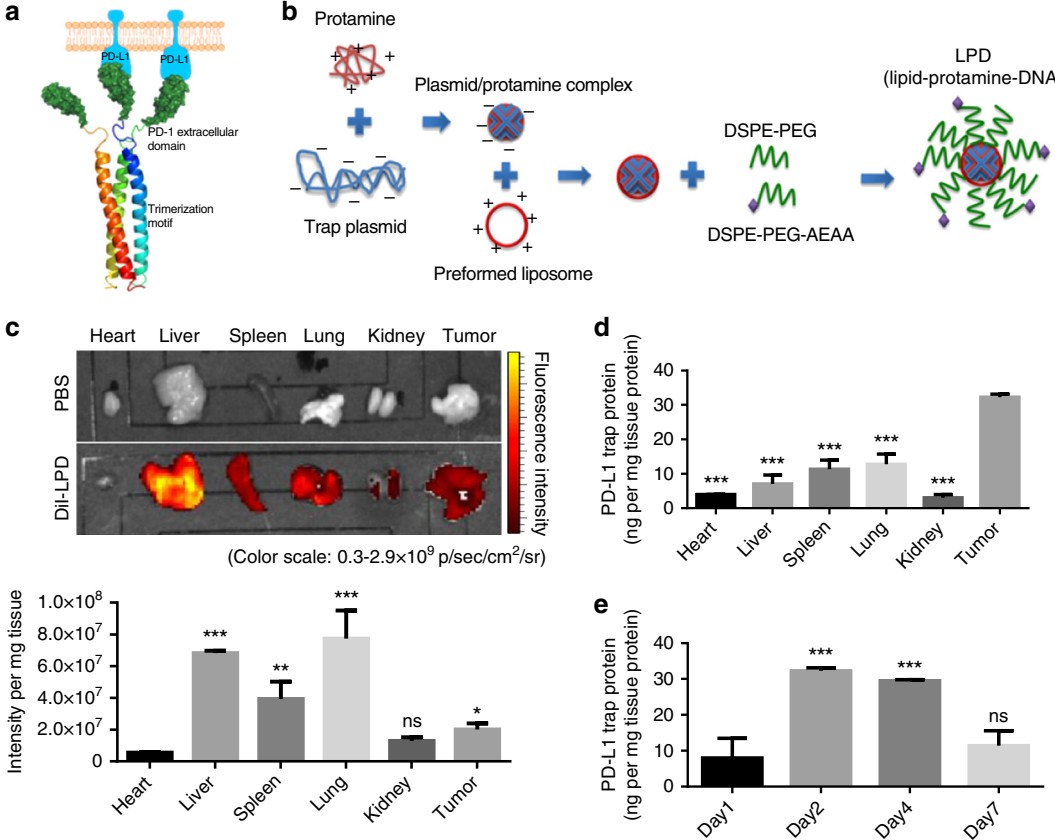

**Fig. 4** Design of locally and transiently expressed PD-L1 trap. **a** Scheme showing the tribody interaction of PD-L1 trap protein. **b** Preparation scheme of PD-L1 trap plasmid loaded LPD. **c** Images and quantitative results of the DiI-loaded LPD in major organs and the CT26-FL3 tumor at 24 h after injection ($n = 3$). **d** PD-L1 trap protein expression in major organs and the CT26-FL3 tumor at 48 h after injection. The PD-L1 trap expression was measured using ELISA by targeting the His (6×)-tag engineered at the C-terminus of the PD-L1 trap ($n = 3$). **e** PD-L1 trap protein expression in tumors on days 1, 2, 4 and 7 after injection. Significant differences in **c**, **d**, and **e** were assessed using $t$ test. Results are presented as mean (SD). ns, not significant. *$P < 0.05$, **$P < 0.01$, ***$P < 0.001$

self-assembly, binds to mouse PD-L1 with a $K_d$ of 219 pM, a binding affinity more than a thousand times higher than that between monomeric PD-1 and PD-L1[24,25]. This high affinity allows to efficiently disrupt the otherwise extensive cell surface interactions between endogenous PD-1 on T-cells and PD-L1 on cancer cells. The optimized coding sequence (Supplementary Note 1) for the secreted form of trap protein was cloned into the expression vector pcDNA3.1, and the PD-L1 trap plasmid was encapsulated into a LPD nanoparticle system, following a previously published protocol[26–28]. In brief, PD-L1 trap plasmid was condensed with cationic protamine to form a slightly anionic complex core, which was further coated with the preformed cationic liposomes (DOTAP and cholesterol), and modified with DSPE-PEG and tumor targeting DSPE-PEG-AEAA (Fig. 4b). The LPD nanoparticles thus formed have a hydrodynamic diameter of ~129 nm and surface charge of ~44 mV (Supplementary Fig. 5). After injection into orthotopic CT26-FL3 tumor-bearing mice through tail vein, it was found that the liver, spleen, lung, and tumor were the major LPD accumulation sites (Fig. 4c). Significantly, the major expression of PD-L1 trap protein was observed in the tumor, presumably due to a combination of the enhanced permeability and retention (EPR) effect and the AEAA-mediated targeting effects resulting in more efficient transfection of the trap plasmid in the tumor tissue (Fig. 4d). Importantly, the expression of the PD-L1 trap protein is transient as we expected, with the highest expression from day 2 to day 4, and significantly reduced expression on day 7 (Fig. 4e, Supplementary Fig. 6). Thus, through the nanoparticle-based gene delivery and tumor-preferred expression, the LPD-PD-L1 trap plasmid system provides a means for selective PD-L1 blockade in the tumor microenvironment.

**Combination of OxP and PD-L1 trap**. Through our findings, we hypothesize that a combination of OxP and locally expressed PD-L1 trap for orthotopic CT26-FL3 tumor therapy may result in synergistic effect with low adverse effects. As shown before, the OxP treatment turned the "cold" tumor into "hot", while PD-L1 blockade released the immune restriction by high levels of checkpoint inhibitory proteins and immunosuppressive cytokines in the tumor. Presumably, locally expressed PD-L1 trap will not release immune restrictions in normal physiological processes and may reduce unnecessary irAEs. To test this hypothesis, we gave the orthotopic CT26-FL3 tumor-bearing mice two injections of OxP and three injections of LPD-PD-L1 trap plasmid (PD-L1 trap for short in following statements) according to the treatment scheme illustrated in Fig. 5a. Significantly, this combination treatment resulted in much efficient tumor inhibitory effect (Fig. 5b, Supplementary Fig. 7). Both CD4[+] and CD8[+] T cells played important roles in the combination therapy, as their depletion using anti-CD4 or anti-CD8 mAb significantly diminished the therapeutic efficacy (Supplementary Fig. 8). In comparison with PD-L1 trap alone or OxP alone treatment, the combination of OxP and PD-L1 trap resulted in significantly elevated TSR% on day 35 and much prolonged survival time as demonstrated in Fig. 5c, d, respectively. Besides, OxP+PD-L1 trap showed better tumor inhibition effect than OxP +anti-PD-L1 mAb (Fig. 5c), presumably due to the locally expression and prolonged secretion of PD-L1 trap inside the tumor tissue. We further examined the immune microenvironment of the tumor after the combination therapy. Similar to OxP treatment, CD8[+] T cells, CD4[+] T cells and activated DCs were significantly increased inside the tumor, while major changes in PD-L1 level was

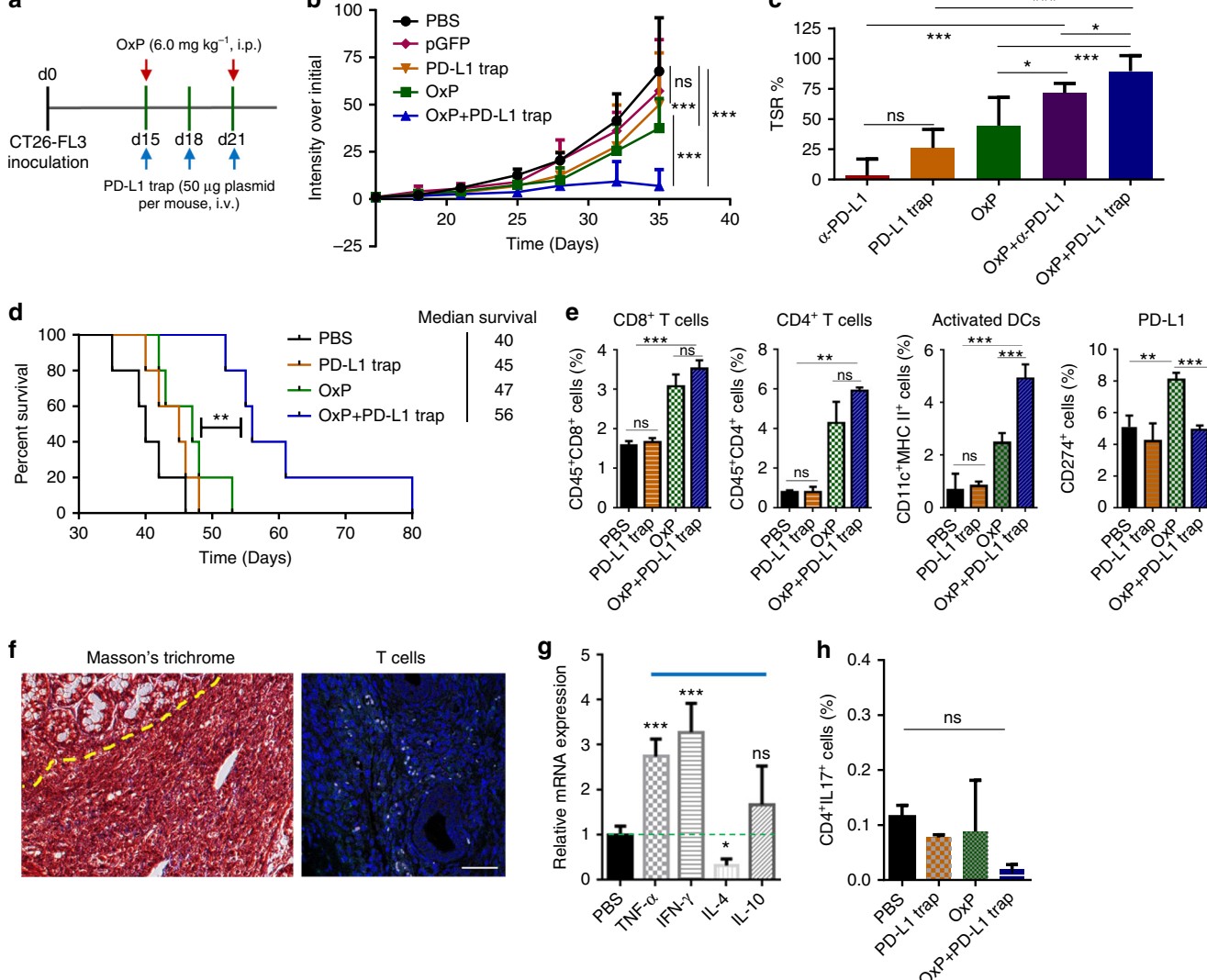

**Fig. 5** Combination of OxP and LPD-PD-L1 trap gene therapy on orthotopic CT26-FL3 tumor model. **a** OxP and PD-L1 trap combination treatment scheme. **b** Tumor growth curves of orthotopic CT26-FL3 tumors in PBS, LPD-pGFP, PD-L1 trap, OxP and OxP+PD-L1 trap gene treated groups ($n = 5$ mice per group). **c** TSR% results on day 35. **d** Mice survival curves. **e** CD8[+] T cells, CD4[+] T cells, activated DCs and PD-L1 levels in tumors of mice after various treatments, analyzed by flow cytometry ($n = 4$). **f** Masson's trichrome and immunofluorescence staining of the orthotopic tumors after OxP+PD-L1 trap treatment using DAPI (blue) and anti-CD3 (red). Yellow dotted line indicates the border between intestinal mucosa and the orthotopic tumor. Scale bar represents 50 μm. **g** Relative mRNA expressions of cytokines in tumors of the OxP+PD-L1 trap treated group compared to PBS group on day 28, detected by quantitative RT-PCR ($n = 4$). **h** Th17 cell ratios in the splenocytes of mice after various treatments on day 28 ($n = 4$). Significant differences were assessed in **b** using two-way ANOVA with multiple comparisons, in **d** using log rank test and in **c**, **e**, **g**, and **h** using $t$ test. Results are presented as mean (SD). ns, not significant. $*P < 0.05$, $**P < 0.01$, $***P < 0.001$

not observed (Fig. 5e). Importantly, T cells were found to be dispersed throughout the tumor tissue (Fig. 5f), and abundant apoptosis was observed in tumors endured the combination therapy (Supplementary Fig. 9). The Th1-type cytokines IFN-γ and TNF-α levels were increased, while the changes of IL-4 and IL-10 levels were not significant compared to the PBS group (Fig. 5g). These data confirmed that the combination of OxP and PD-L1 trap revoked the immunosuppressive tumor microenvironment of orthotopic CT26-FL3 tumor, resulting in the activation of T cells and consequently increased efficacy of the immune therapy.

We further checked the Th17 cells in the spleens after various treatments. Unlike what was observed in the anti-PD-L1 mAb and OxP+anti-PD-L1 mAb treatment groups (Fig. 3e), neither the PD-L1 trap alone nor the OxP+PD-L1 trap treatment groups showed any significant increases of the Th17 cells in the spleen (Fig. 5h). These results further justified the low toxicity and low

adverse effects of our local and transient expression of PD-L1 trap fusion protein strategy compared to the conventional systemic administration of anti-PD-L1 mAb. Indeed, the administration of OxP+PD-L1 trap induced neither obvious loss of bodyweight (Supplementary Fig. 10), nor abnormal changes of the complete blood count and blood chemistry (Supplementary Fig. 11). Abnormality of histological structure of spleen, heart, lung and kidney was not observed (Supplementary Fig. 12). It should be noted that while spontaneous liver metastasis was observed in the PBS and PD-L1 trap group, no metastasis was observed in the OxP and combination group (Supplementary Fig. 12). These results strongly indicate that the combination of low dose OxP and local and transient expression of PD-L1 trap fusion protein is an effective and low adverse effect strategy for the treatment of MSS/MMR-proficient colorectal cancer.

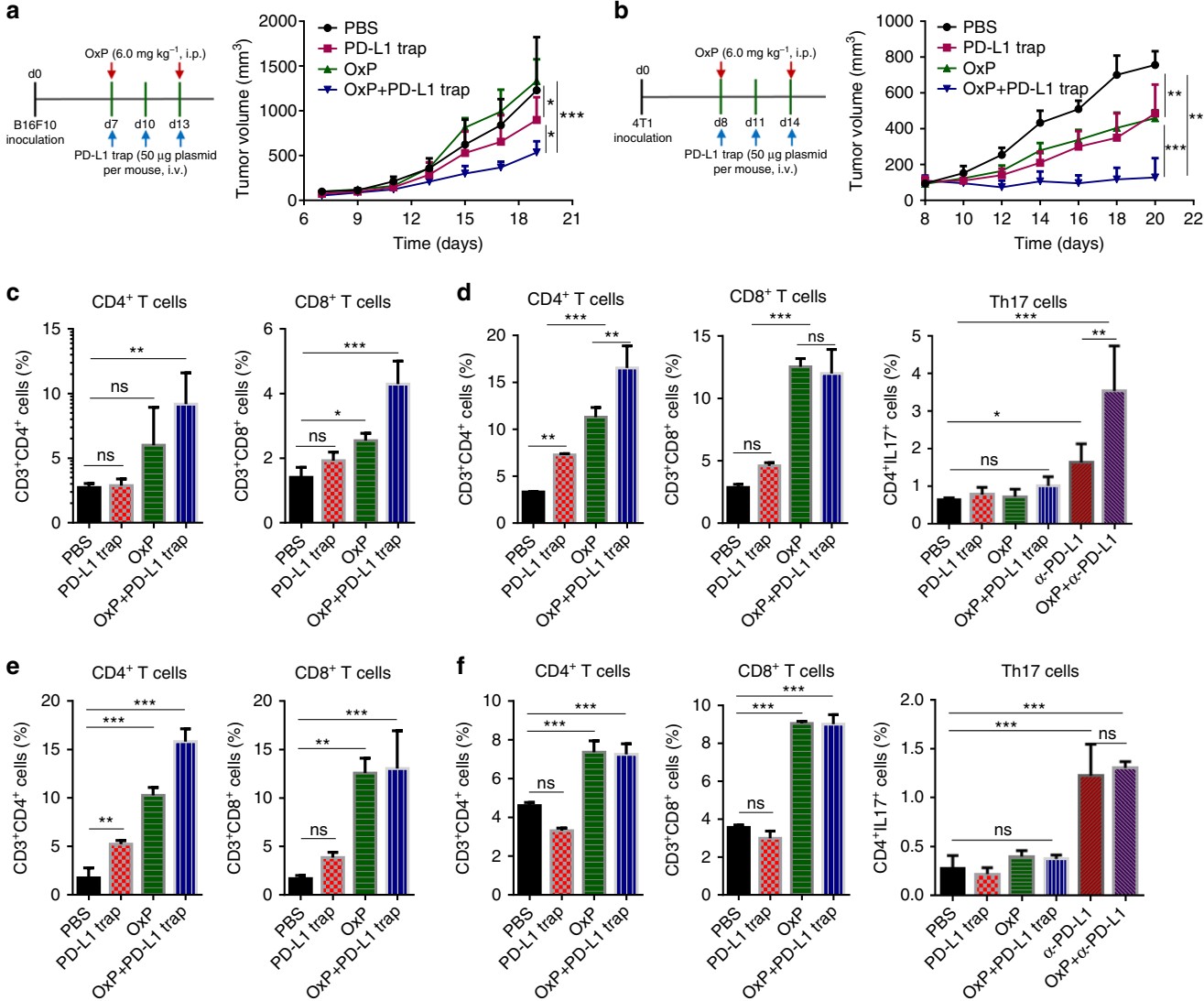

**Fig. 6** OxP and LPD-PD-L1 trap gene therapy on B16F10 and 4T1 tumor models. **a** Treatment scheme and tumor growth curves of B16F10 tumors in PBS, PD-L1 trap, OxP and OxP + PD-L1 trap treated groups ($n = 5$ mice per group). **b** Treatment scheme and tumor growth curves of 4T1 tumors in PBS, PD-L1 trap, OxP and OxP + PD-L1 trap treated groups ($n = 5$ mice per group). **c** CD4+ and CD8+ T cell ratios in the B16F10 tumors after various treatments, analyzed by flow cytometry ($n = 4$). **d** CD4+ T cell, CD8+ T cell and Th17 cell ratios in splenocytes of the B16F10 tumor-bearing mice after various treatments, analyzed by flow cytometry ($n = 4$). **e** CD4+ and CD8+ T cell ratios in the 4T1 tumors after various treatments, analyzed by flow cytometry ($n = 4$). **f** CD4+ T cell, CD8+ T cell and Th17 cell ratios in splenocytes of the 4T1 tumor-bearing mice after various treatments, analyzed by flow cytometry ($n = 4$). Significant differences were assessed in **a** and **b** using two-way ANOVA with multiple comparisons and in **c**, **d**, **e**, and **f** using $t$ test. Results are presented as mean (SD). ns, not significant. $^*P < 0.05$, $^{**}P < 0.01$, $^{***}P < 0.001$

**Test in other MMR-proficient models**. We further tested the effect of combination of OxP and PD-L1 trap on two other non-hypermutated MMR-proficient murine cancer models. Murine melanoma model was established by subcutaneous injection of B16F10 cells into C57BL/6 mice, and metastatic breast cancer model was established by injection of 4T1 cells into the breast pad of BALB/c mice. B16F10 and 4T1 cell lines were both originally obtained from spontaneous tumors. Previous reports have confirmed that these two murine cell lines are both weakly immunogenic, with much less single nucleotide point mutations than other murine tumors, and no mutations in *Tp53*, *Braf*, *Pold1*, and MMR genes *Mlh1*, *Msh2*, *Msh6*, and *Pms2*[29–31]. As observed in orthotopic CT26-FL3 tumors, both B16F10 and 4T1 tumors were refractory to PD-L1 blockade therapy, while the combination of OxP and PD-L1 trap efficiently reduced the tumor burden (Fig. 6a, b, Supplementary Fig. 13). Further analysis showed elevated CD4+ and CD8+ T cell ratios in the tumor, as well as in

the spleen after OxP or OxP+PD-L1 trap treatment in both models (Fig. 6c–f), and confirmed the effect of OxP in promoting anti-tumor responses. In contrast to anti-PD-L1 mAb treatment, there were no obvious Th17 cell populations observed in the spleens after the PD-L1 trap or OxP+PD-L1 trap treatments in both models (Fig. 6e, f), confirming the low tendency in inducing autoimmune syndromes by this therapy.

**Colorectal cancer patient samples analysis**. OxP is among the first-line chemotherapy of colorectal cancer, and this provides a possibility of whether colorectal tumors after OxP-based chemotherapy make a good condition for PD-1/PD-L1 inhibitor therapy. We assessed hot spot analysis and immunofluorescence staining in tumor biopsy sections from MSS colorectal cancer patients who were genotyped for lack of mutations in *Mlh1*, *Msh2*, *Msh6*, and *Pms2*. In consistent with the results observed in

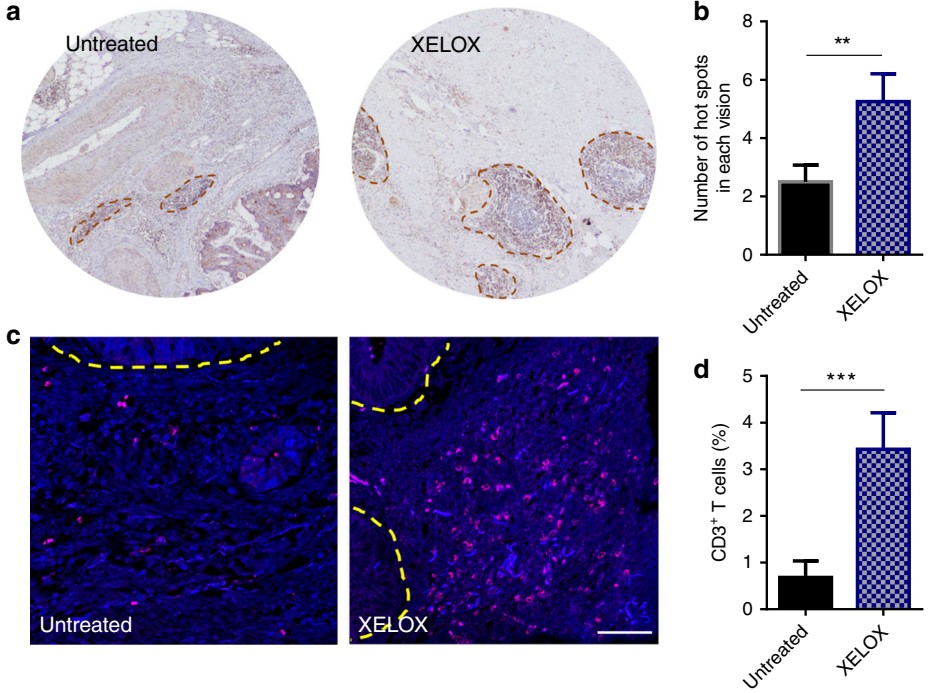

**Fig. 7** MSS colorectal cancer patient sample analysis. **a** Representative immunohistochemistry images of tumor tissues from MSS colorectal cancer patients. Untreated represents samples from patients without any therapy. XELOX represents samples from patients after XELOX neoadjuvant chemotherapy. Images are taken at ×40 magnification. Dark red circles indicate the hot spots in each visual field. **b** Statistics of hot spots in each visual field ($n = 4$ samples per group). **c** Representative immunofluorescence images of tumor tissues using DAPI (blue) and anti-CD3 (red). Yellow dotted line indicates the border between intestinal mucosa and the tumor. Scale bar represents 50 µm. **d** Statistics of CD3$^+$ T cell ratios in each slice ($n = 5$ samples per group). Significant differences were assessed in **b** and **d** using $t$ test. Results are presented as mean (SD). \*\*$P < 0.01$, \*\*\*$P < 0.001$

the orthotopic CT26-FL3 tumor model, more hot spots were observed in the colorectal tumors from patients endured 2–3 courses of XELOX (capecitabine plus OxP) treatment, compared to those from patients without any treatment (Fig. 7a, b). Furthermore, we analyzed 10 patients for T-cell infiltration in the tumor by immunofluorescence staining. Similarly, profound T cells were observed in the tumors from patients endured XELOX treatment, while much less T cells were seen in the tumor from patients without any treatment (Fig. 7c, d). This result confirmed that OxP-based neoadjuvant chemotherapy may provide a good condition for application of PD-1/PD-L1 blockade, especially locally expressed PD-L1 trap, in MSS colorectal cancer.

## Discussion

Checkpoint blockade immunotherapies are being extensively investigated for the treatment of various malignant tumors. A major challenge in the field is that durable antitumor responses and long-term remissions are only demonstrated in limited cancer types. Currently, there is an urgent need in developing innovative combination therapy to broaden the cancer type responses to PD-1/PD-L1 blockade. Several biomarkers, including high levels of PD-L1 expression, Th1-type chemokines, infiltrating T cells, somatic-mutations, and low levels of immunosuppressive elements have been confirmed to be associated with an active response to PD-1/PD-L1 blockade[32,33]. Recent reports showed that combination of anti-PD-1/PD-L1 therapy with MEK inhibitor[34], BTK inhibitor[35], PPARγ/RXRα inhibitor[36], TNF superfamily member LIGHT[37,38], immunogenic chemotherapy[39], photodynamic therapy[40,41], and irradiation[42,43] would promote T-cell infiltration and anti-tumor activity of checkpoint inhibitors. These studies highlight the importance of reviving the immunosurveillance in the tumor microenvironment in sensitize tumors to PD-1/PD-L1 blockade therapy.

In this work, we focus our study on non-hypermutated MSS/MMR-proficient cancer types. Clinical results of pembrolizumab showed that hypermutated MSI-H/MMR-deficient tumors were more responsive to PD-1/PD-L1 blockade therapy, while non-hypermutated MSS/MMR-proficient tumors showed almost no responses. We established a syngenic, orthotopic colorectal mouse cancer model with MMR-proficient CT26-FL3 cells, and proved that administration of a relative low dose of OxP to the CT26-FL3 tumor-bearing mice resulted in a series of immunogenic effects inside the tumor, and significantly boosted effect of PD-L1 blockade in tumor burden control. OxP was applied here as an ICD inducing agent because it is among the first-line chemotherapy protocols for colorectal cancer as well as many other cancer types. In a previous report, Shalapour et al.[44] showed that OxP treatment increased CXCL13 levels in prostate cancer and recruited immunosuppressive plasma cells to the tumor, which would impede the T-cell-dependent immunogenic chemotherapy. In this study, we also observed a slight increase in CXCL13 mRNA expression in the OxP treatment group, while no obvious PD-L1$^+$IL-10$^+$ plasma cell population was observed, suggesting different responses among different animal models. However, ELSs were found in the OxP-treated CT26-FL3 tumor model, with abundant T cells, DCs, macrophages, NK cells, Tregs, MDSCs, as well as increased expression of immunosuppressive PD-L1 and IL-10. The elevated expression of PD-L1 protein elucidated the necessity of combining OxP-based immunogenic chemotherapy with PD-L1 blockade in orthotopic CT26-FL3 tumor treatment.

A locally and transiently expressed PD-L1 trap was applied in this work for PD-L1 blockade. Immune checkpoint is involved in the maintenance of immunologic homeostasis and helping to maintain peripheral tolerance of self-molecules to prevent excess autoimmunity. Systemically applied mAb that block these

immune checkpoint molecules may disrupt the balance in immunologic tolerance and lead to autoimmune-like/inflammatory side-effects in normal organ systems or tissues[45,46], including type 1 diabetes and heart attack as recently reported[47]. Local and transient expression of checkpoint inhibitors in tumor microenvironment provides an ideal option to reduce these irAEs. In the study here, the combination of OxP and locally expressed PD-L1 trap did not induce appearance of Th17 cells in the spleens as observed in the anti-PD-L1 mAb treated mice in all three models, indicating our strategy is a more efficient and safer option for cancer immunotherapy.

To confirm the clinical relevance of this study, we analyzed colorectal cancer patient samples genotyped as MSS type. In the patients treated with OxP-based neoadjuvant chemotherapy (XELOX), more T-cell infiltration was observed in the tumor. These results support further investigation of the combination use of PD-L1 inhibitor with OxP-based chemotherapy in MSS colorectal cancer patients. Specifically, several points need clarification in further clinical studies: (1) What is the proper time course for applying PD-L1 inhibitor and OxP-based chemotherapy? In a pilot study, stage III colon cancer patients receiving standard OxP/capecitabine chemotherapy were vaccinated at the same time with keyhole limpet hemocyanin (KLH) and carcinoembryonic antigen (CEA)-peptide pulsed DCs, and enhanced T-cell reactivity upon OxP administration was observed[48]. However, results of pembrolizumab were still not impressive in MSS colorectal cancer patients who have failed in previous chemotherapy. Should PD-L1 inhibitor be used as a first-line treatment together with OxP-based chemotherapy, like that applied in metastatic non-squamous non-small cell lung cancer? (2) Dosage is an important factor and should be carefully tuned in the combination of chemotherapy with immunotherapy. High dose of chemo drugs used in clinical regimens may impair the immune system, and further affect the efficiency of immunotherapy. (3) The tumor mutation burden is still an important criterion for successful immunotherapy. Although artificially boosting the availability of tumor antigens and specific damage-associated molecular patterns by immunogenic chemotherapy efficiently converts non-immunogenic forms of cell demise into instances of ICD, effective anti-tumor immune responses still rely on the availability of antigens—especially neo-antigens—in the tumor.

## Methods

**Materials.** N-(Methoxypolyethylene oxycarbonyl)-1,2-distearoryl-sn-glycero-3-phosphoethanolamine (DSPE-PEG) was purchased from NOF Corporation (SUNBRIGHT® DSPE-020CN). N-(2-aminoethyl)-4-methoxybenzamide conjugated DSPE-PEG (DSPE-PEG-AEAA) was synthesized according to a previously established protocol[49]. 1,2-dioleoyl-3-trimethylammonium-propane chloride salt (DOTAP) was purchased from Avanti Polar Lipids, Inc. Cholesterol and protamine were purchased from Sigma-Aldrich. Oxaliplatin was purchased from Selleckchem. DiI was purchased from ThermoFisher Scientific. β-galactosidase (β-Gal) peptide (TPHPARIGL) and AH1 peptide (SPSYVYHQ) were ordered from Peptide2.0. All other chemicals were purchased from Sigma-Aldrich unless specifically mentioned.

**Cell lines.** Murine colorectal cancer CT26-FL3 cells stably expressing red fluorescent protein (RFP)/Luc were established by transfection of CT26-FL3 cells with vectors carrying RFP/Luc and puromycin resistance gene. The original CT26-FL3 cells were kindly provided by Dr. Maria Pena at the University of South Carolina. Murine colorectal cancer MC38 cells stably expressing green fluorescent protein (GFP)/Luc were established by transfection of MC38 cells with vectors carrying GFP/Luc and puromycin resistance gene. The original MC38 cells were kindly provided by Dr. James J. Moon at the University of Michigan. CT26-FL3 and MC38 cells were cultivated in Dulbecco's Modified Eagle's Medium (DMEM, high glucose, Gibco) supplemented with 10% fetal bovine serum (Gibco), 1% antibiotic-antimycotic (Gibco) and 1 μg mL$^{-1}$ puromycin (ThermoFisher) at 37 °C and 5% CO$_2$ in a humidified atmosphere. Murine melanoma B16F10 cells and murine breast cancer 4T1 cells were obtained from Tissue Culture Facility—UNC Lineberger Comprehensive Cancer Center. B16F10 cells were cultivated in DMEM supplemented with 10% fetal bovine serum and 1% antibiotic-antimycotic at 37 °C and 5% CO$_2$ in a humidified atmosphere. 4T1 cells were cultivated in RPMI 1640 medium supplemented with 10% fetal bovine serum and 1% antibiotic-antimycotic at 37 °C and 5% CO$_2$ in a humidified atmosphere.

**Mouse model establishment.** Six-week-old female BALB/c and C57BL/6 mice were obtained from Charles River Laboratories. All animal handling procedures were approved by the University of North Carolina at Chapel Hill's Institutional Animal Care and Use Committee. Orthotopic CT26-FL3 and MC38 colorectal tumor model was established in female BALB/c and C57BL/6 mice, respectively. Mice were anesthetized by 2.5% isoflurane and placed in supine position. A midline incision was made to exteriorize the cecum. Using an insulin-gauge syringe, $1.0 \times 10^6$ CT26-FL3 or MC38 cells stably expressing luciferase in 100 μL PBS were injected into the cecum wall. Light pressure was applied to the injection site to prevent any leakage. The cecum was returned to the peritoneal cavity. The tumor burden was monitored by intraperitoneal (i.p.) injection of 100 μL of D-luciferin (Pierce™, 10 mg ml$^{-1}$) followed by bioluminescent analysis using an IVIS® Kinetics Optical System (Perkin Elmer, CA). Subcutaneous B16F10 melanoma model was established by injection of $1.0 \times 10^6$ B16F10 cells in 100 μL PBS into the right flank of female C57BL/6 mice. Orthotopic 4T1 breast tumor model was established by injection of $1.0 \times 10^6$ 4T1 cells in 50 μL PBS into the breast pad of female BALB/c mice.

**Tumor growth inhibition assay.** Mice bearing tumors were randomized blindly into different treatment groups, and the investigator was blinded to the group allocation during the animal experiments. For outcome assessment, same protocol was applied across experimental groups. PBS, OxP (6.0 mg kg$^{-1}$, i.p.), anti-mouse PD-L1 mAb (α-PD-L1, Bioxcell, clone 10 F.9G2, 100 μg per mouse, i.p.), LPD-GFP plasmid (pGFP, 50 μg plasmid per mouse, i.v.), LPD-PD-L1 trap plasmid (PD-L1 trap, 50 μg plasmid per mouse, i.v.), OxP + α-PD-L1, OxP + PD-L1 trap, anti-mouse CD8α (α-CD8, Bioxcell, clone 53-6.72, 200 μg per mouse, i.p.) or anti-mouse CD4 (α-CD4, Bioxcell, clone GK1.5, 200 μg per mouse, i.p.) were given at respective schedules. CT26-FL3 and MC38 tumor burdens were monitored using IVIS system every three days. The increases of tumor volumes were calculated as luminescence intensities (photon sec$^{-1}$) over the initial. Tumor suppression rate (TSR%) = $(I_c - I_x)/I_c \times 100\%$, where $I_c$ represents the luminescence intensity of the PBS group, $I_x$ represents the luminescence intensity of other therapy groups. B16F10 and 4T1 tumor diameters were measured by caliper, and the tumor volume ($V_t$, mm$^3$) was calculated by $V_t = 0.5 \times a \times b^2$, where $a$ was the long axis and $b$ was the short axis. TSR% = $(V_c - V_t)/V_c \times 100\%$, where $V_c$ and $V_t$ are the tumor volumes of the PBS and other treatment groups. Bodyweight was recorded every week. At designed time points, 4–5 mice in each group were sacrificed. Tumors were harvested for Masson's trichrome staining, immunofluorescence staining, flow cytometry analysis and RT-PCR assay. Spleens were harvested for flow cytometry analysis.

**Immunofluorescence staining.** Immunohistochemistry was performed on paraffin-embedded sections from tumor tissues. All tissues for paraffin-embedding were resected, rinsed in PBS, and placed in 4% PFA for over 48 h at 4 °C. Immunofluorescence staining of paraffin-embedded sections was performed by deparaffinization, antigen retrieval, permeabilization, and blocking in 5% goat serum at room temperature for 1 h. Primary antibodies were incubated at 4 °C overnight, and fluorescent secondary antibodies were incubated at 37 °C for 1 h. Finally, the slices were mounted with Prolong® Diamond Antifade Mountant with DAPI (ThermoFisher Scientific). Primary antibodies, fluorescent primary and secondary antibodies used for immunofluorescence staining were listed in Supplementary Table 1. Immunohistochemistry images were taken on a microscope (Nikon Eclipse Ti-U). Immunofluorescence images were taken on a laser scanning confocal microscope (Zeiss LSM 700).

**CRT exposure and HMGB1 release test.** For surface detection of CRT, CT26-FL3 cells were incubated with OxP for 4 h at a concentration of 25 μM. After that, cells were placed on ice, washed twice with PBS and fixed in 0.25% PFA in PBS for 5 min. Cells were washed twice in PBS, and a primary antibody, diluted in cold blocking buffer, was added and incubated for 30 min. After three washes with cold PBS, cells were incubated for 30 min with the appropriate secondary antibody diluted in a cold blocking buffer. Cells were fixed with 4% PFA for 20 min and then mounted with Prolong® Diamond Antifade Mountant with DAPI. For intracellular HMGB1 staining, CT26-FL3 cells were incubated with OxP for 24 h at a concentration of 300 μM. After that, cells were washed with PBS, fixed with 4% PFA for 20 min, permeabilized with 0.1% Triton X-100 for 10 min, rinsed three times with PBS, and nonspecific binding sites were blocked with 5% goat serum in PBS for 30 min. A primary antibody was added and incubated for 1 h. Subsequently, cells were washed three times with PBS and incubated for 30 min with an appropriate secondary antibody. Primary antibodies, fluorescent primary and secondary antibodies used for immunofluorescence staining were listed in Supplementary Table 1.

For quantitative determination of the HMGB1 release, CT26-FL3 cells were seeded in 6-well plates with 1 mL full medium. The medium was changed 24 h later and different concentrations of OxP were added to the cells. 24 h later, the supernatants were collected, dying tumor cells were removed by centrifugation, and

supernatants were isolated and frozen immediately. Quantification of HMGB1 in the supernatants was assessed by mouse HMGB1 ELISA Kit (LS-F11642, LifeSpan BioSciences, Inc.) according to the manufacturer's instruction.

**ELISpot assay for IFN-γ production**. A total of $1 \times 10^6$ CT26-FL3 cells were treated with medium alone or OxP (25 μM) for 4 h, harvested and injected into BALB/c mice subcutaneously. Seven days later, the spleens were harvested and made into single-cell suspensions in a sterile cell-culture hood. Cells were seeded at $2 \times 10^5$ cells per well in a capture antibody coated 96-well plate. The single-cell suspensions were then co-cultured with 15 μg mL$^{-1}$ of β-Gal or AH1 peptide at 37 °C for 18 h. Cells were subsequently removed by several wash steps. The production of IFN-γ was measured by adding detection antibody, followed by enzyme conjugate magnification. Brown dots signals were developed with a BD$^{TM}$ ELI-SPOT AEC substrate set and calculated manually. For splenocytes from tumor-bearing mice treated with PBS or OxP, similar procedure was applied.

**Anti-tumor vaccination**. A total of $1 \times 10^6$ CT26-FL3 cells were treated with medium alone or OxP (25 μM) for 4 h, collected and injected into BALB/c mice subcutaneously. After 7 days, $5 \times 10^5$ CT26-FL3 cells were inoculated into the contralateral flank. Tumor occurrence was recorded and tumor volume was measured every 2–3 days. Tumor volume ($V_t$) was calculated according to the following equation: $V_t = a \times b^2/2$, where $a$ and $b$ are the major and minor axes of the tumor, respectively.

**Flow cytometry assay**. Spleen and tumor tissues were made into single-cell suspensions, and the splenocytes and tumor-infiltrating lymphocytes were quantitatively analyzed by flow cytometry after immunofluorescence staining. In brief, tissues were harvested and digested with collagenase A and DNAase at 37 °C for 40–50 min. After red blood cell lysis via addition of ACK buffer, cells were collected and dispersed with 1 mL of PBS, and stained by the addition of a cocktail of fluorescence conjugated antibodies. Following staining, cells were fixed with 4% PFA and analyzed via FACS (BD LSR II). Fluorescence conjugated antibodies used for flow cytometry are listed in Supplementary Table 1.

**Quantitative real-time PCR (qPCR) assay**. Total RNA was extracted from the tumor tissues using an RNeasy® Microarray Tissue Mini Kit (Qiagen). cDNA was reverse-transcribed using the iScript$^{TM}$ cDNA Synthesis Kit (BIO-RAD). In total 150 ng of cDNA was amplified with the TaqMan$^{TM}$ Gene Expression Master Mix. All the mouse-specific primers for RT-PCR reactions are listed in Supplementary Table 2. GAPDH was used as the endogenous control. Reactions were conducted using the 7500 Real-Time PCR System and the data were analyzed with the 7500 Software.

**Preparation and characterization of LPD nanoparticles**. PD-L1 trap plasmid was constructed by assembling the coding sequences of the PD-1 extracellular domain (mouse PD-1 residues 21–150) and the C-terminal trimerization domain of cartilage matrix protein (mouse CMP1 residues 458–500), with a flexible hinge region and a sequence cording for a secretion signaling peptide, cloned into pcDNA3.1 between Nhe I and Xho I sites[28]. LPD nanoparticles loaded with GFP plasmid or PD-L1 trap plasmid were prepared through a stepwise self-assembly process based on a well-established protocol[50]. Briefly, DOTAP/cholesterol liposomes were prepared by the hydration-extrusion method. LPD polyplex cores were formulated by mixing of 20 μg protamine in 100 μL DI water with 50 μg plasmid in equal volume of DI water. The mixture was incubated at room temperature for 10 min and then 60 μL of the pre-formed cholesterol/DOTAP liposomes were added. Post insertion of 10 μL DSPE-PEG and 10 μL DSPE-PEG-AEAA was further performed at 60 °C for 15 min. Finally, 20 μL 20% glucose solution was added to adjust the osmotic pressure. The size and surface charge of the nanoparticles were determined by a Malvern ZetaSizer Nano series (Westborough, MA). Transmission electron microscopy (TEM, JEOL 1230) images were acquired where nanoparticles were negatively stained with 2% phosphotungstic acid (PTA).

**Biodistribution of LPD nanoparticles**. Approximately 0.05% of hydrophobic dye DiI was incorporated into DOTAP liposomes to formulate the DiI-labeled LPD nanoparticles. After 24 h of intravenous injection of the DiI-labeled LPD nanoparticles, mice were killed and major organs and tumors were collected. The distribution of LPD nanoparticles in major organs was quantitatively visualized with IVIS system, with the excitation wavelength at 520 nm and the emission wavelength at 560 nm.

**In vivo expression of PD-L1 trap protein and GFP**. LPD nanoparticles encapsulated with PD-L1 trap plasmid were intravenously injected (50 μg plasmid per mouse) into CT26-FL3 orthotopic tumor-bearing mice. At day 1, 2, 4, and 7 post-injection, mice were sacrificed, major organs and tumors were collected and homogenized in the RIPA buffer. Total protein concentration in the lysate was determined through a Pierce$^{TM}$ BCA Protein Assay Kit. The amount of PD-L1 trap expressed was determined by detecting the His-Tag using a His-Tag ELISA detection kit (GenScript).

In another group, mice bearing orthotopic CT26-FL3 tumor were intravenously injected with LPD nanoparticles encapsulating GFP plasmid. At day 1, 2, 4, and 7 post-injection, tumor tissues were cryo-sectioned. GFP protein expression images were taken on Zeiss LSM 700.

**TUNEL assay**. TUNEL assays were carried out using a DeadEnd$^{TM}$ Fluorometric TUNEL System (Promega) following the manufacturer's instruction. Fragmented DNA of apoptotic cells were fluorescently stained with FITC (green) and defined as TUNEL-positive nuclei. Slides were cover-slipped with Prolong® Diamond Anti-fade Mountant with DAPI. Images were taken using CLSM.

**Blood chemistry analysis**. On day 28, a week after the final treatments, three mice in each group were subjected to a toxicity assay. Both whole blood and serum were collected. Whole blood cellular components were counted and compared. Creatinine (CRE), blood urea nitrogen (BUN), serum aspartate aminotransferase (AST), and alanine aminotransferase (ALT) in the serum were assayed as indicators of renal and liver functions. Organs including the heart, liver, spleen, lungs, and kidneys were collected and fixed for hematoxylin and eosin (H&E) staining at UNC histology facility to evaluate the organ-specific toxicity and spontaneous metastasis.

**Colorectal cancer patient tumor samples**. Sections from paraffin-embedded biopsies of colon resections from colon cancer patients were obtained from the Department of Gastrointestinal Surgery at the Second Hospital of Jilin University and approved by the ethics committee. These tumors were known to bear no mutations in *Mlh1*, *Msh2*, *Msh6*, and *Pms2*, indicating these samples all belong to MSS type colon cancer. Among them, nine samples are from patients without any therapy, and nine samples are from patients who have endured 2–3 courses of neoadjuvant chemotherapy of XELOX (capecitabine plus OxP). Immunohistochemistry and immunofluorescence were performed and evaluated blindly based on a defined scoring method.

**Statistical analysis**. To compare between two groups, unpaired two-tailed $t$ test was used. For comparison between multiple groups, ordinary two-way ANOVA with multiple comparisons adjusted by Šidák correction was used. For survival analyzes, log rank test was used for comparison. All statistical analysis was performed using Prism 6.0 Software. Appropriate tests were applied in analyzing these data, meeting assumptions of the statistical methods. No exclusion criteria were incorporated in the design of the experiments for this study. For animal experiments, a sample size of 5 was chosen for each experimental group. We can detect an effect between our groups of interest with a statistical power greater than 0.8, under a confidence level of 95% with our chosen sample size.

**Data availability**. All data generated or analyzed during this study are included in this published article (and its Supplementary Information Files) or available from the corresponding author on reasonable request.

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

## Acknowledgements

This work was supported by NIH grants CA198999 (to L.H.) and CA157738 (to R.L.) and a grant from Eshelman Institute for Innovation (to L.H. and R. L.). It was also supported by NSFC Project 51673185 (to W.S.) and China Scholarship Council.

## Author contributions

W.S., R.L., and L.H. designed the project, analyzed data and wrote the manuscript. W.S., Y.W., and Q.L. prepared the nanoparticles, performed the in vitro experiments. W.S., Y. W., and T.J.G. performed the surgery and in vivo experiments. L.S. performed the flow cytometry test. J.L. and O.D. generated the trap plasmid. T.L. provided patient tumor samples and analysis. All authors discussed the results and commented on the manuscript.

## Additional information

**Competing interests:** The trap technology has been licensed to OncoTrap Inc., L.H. and R.L. are co-founders. The remaining authors declare no competing interests.

