## [Peer Review File · Nature Communications]

Reviewers' comments:

Reviewer #1 (Remarks to the Author):

This work suggests that oxaliplatin may have the potential of changing the tumor microenvironment from an immune excluded tumor to an inflamed tumor, making it susceptible to immune checkpoint inhibition. Several articles have already shown that OXP can induce immunogenic cell death in this model. The authors of this article characterize the immune infiltrate in CT26 tumors following OXP treatment, and explore the combination with immune checkpoint inhibition of PD-L1 to unleash the OXP-induced antitumor immune response. They show that, indeed, the combination appears to have synergistic antitumor efficacy and this is more evident when they combine OXP with the PD-L1 trap, rather than anti-PD-L1.

Major points:

1. Although the article is nicely written and addresses an important question, the previous articles published regarding the immunogenic cell death induced by OXP partially decrease the novelty of the findings. In addition, the authors do not provide any additional mechanistic data.
2. Another concern is the fact that the authors extrapolate a conclusion for all MSS tumors based on one model, CT26. Although they describe it is poorly immunogenic, this tumor was induced using a carcinogen and, hence, harbors a good number of non-synonymous somatic mutations and potential neoantigens (Castle et al. BMC Genomics 2014, 15:190). Does this model represent well the biology of MSS tumor and its potential immunogenicity, and is one model enough to extrapolate these findings? It would be interesting to see if other poorly immunogenic tumors (B16) could also be rendered susceptible to immune checkpoint inhibition through combination with OXP. This would make the findings more relevant.
3. Immunogenic cell death had already been shown in this model. The authors should reference the previous literature (line 194 and others).
4. Fig. S3b, did the authors test against irrelevant peptide to evaluate whether the response is peptide specific and not just non-specific? Are there internal positive and negative controls to ensure the cells are alive and capable of IFN-gamma secretion?
5. Additional IHC data on human tumor samples examining the expression of effector cytokines, PD-L1 or other markers would provide stronger evidence that OXP induces immunogenic cell death in human CRC MSS samples.
6. How many times were each of the in vivo experiments repeated? How reproducible is the in vivo data?

Minor points:

7. What is the control in Figure S3c? It is not clear in the text nor figures.
8. The statement in line 218: "However, the abundant checkpoint proteins and immunosuppressive cytokines inside the ELSs definitely impaired the biological functions of T cells inside the Oxp treated tumors" is very strong. The data is suggestive, but not conclusive.
9. Fig. 4g and h; It would be interesting to display the values for all the groups, rather than for selected groups, to evaluate the contribution of each of the therapies individually and in combination.
10. The authors make the observation that PD-L1 expression is induced in vivo despite lack of T cell infiltration in CT26. What drives PD-L1 expression if not IFN-gamma? Macrophages? Other cells?

Reviewer #2 (Remarks to the Author):

In their manuscript, Dr. Song and colleagues investigate the consequences of oxaliplatin-mediated sensitization to immune checkpoint treatment. The authors show that in a colorectal murine cell lines (CT26) the anti PD-L1 treatment does not affect a host-mediated immune-control. The

opposite was obtained when they combine oxaliplatin and PD-L1 modulator owing to an HMGB1 mediated immunogenic cell death. In principle these are interesting findings, however the preclinical models they used has limitations ad discussed below. Some of the experiments are not convincing or lack proper controls. Furthermore the implications in the relevant clinical setting (MSS CRC) are not convincing.

The authors define MC38 and CT26 as an MSI-H and MSS cell line respectively. They claim that "mismatch repair genes Msh3 as well as Pold1 were mutated in MC38 cell line, indicating this cell line can be used for a valid model for studying MSI-H colorectal cancer". Although the authors mention a reference to sustain this claim, the paper they reference is unclear and unconvincing (Efremova et al, bioRxiv, 2017). In 2014 Yadav et al published a manuscript in Nature where they genetically characterized MC38. The latest should be used as a reference.

To make the claim that MC38 is MSI, a microsatellite analysis should be performed, this should be coupled to NGS. PCR primers to detect microsatellite shift in mouse cells using the matched mouse background are available. The functional impact of alterations in mismatch repair genes should also be validated.

The amount of somatic mutations (tumor mutational burden) in MC38 and CT26 could greatly impact immune-control and immune surveillance. The authors do not mention the mutational load of CT26 and MC38 that are two chemically induced tumors. The mutational burden of the actual MC38 and CT26 used in this study should be measured by NGS and inserted in the paper.

The authors show tumor shrinkage of MC38 after PD-L1 administration. The consequences of oxaliplatin treatment alone on MC38 are of interest. Have MC38 tumors complete rejection?

An interesting section of the manuscript regards the use of PD-L1 trap plasmid that delivered via liposome-protamine-DNA nanoparticles, locally and transiently, produces PD-L1 blocker in tumor tissue. The in vivo experiments are well conducted but they lack of appropriate controls. Indeed, a pcDNA3.1 backbone should be matched with the PD-L1 trap and in vivo injected. This is could help the authors discriminating for an immune response against the plasmid (exogenous DNA vectors and the proteins they express have ben shown to be immunogenic) and the nanoparticles present in the system.

The authors further show that Oxaliplatin induces Immunogenic cell death in CT26-FL3 Tumor. Though of putative interest, this working hypothesis is not original since other publications, have already reported oxaliplatin mediated immunogenic cell death. Tesniere and Kroemer in an Oncogene paper (2010), which the authors properly acknowledge, described the HMGB1-dependent activation of the immunogenic cell death in CRC post oxaliplatin treatment. Overall the concept that CRC cell lines undergo immunogenic cell death upon oxaliplatin treatment is not novel.

Potentially a most interesting (and translational) part of the manuscript is the one dealing with the effect of the oxaliplatin on MSS CRC patients. In this setting the therapeutic choice is limited and immune-checkpoint blockade have failed, most likely due to the low immunogenicity of MSS CRCs. The authors analyze public databases and make the claim that treatment oxalipl (capecitabine plus OxP infact) affects the status (appearance of ELS structures) of CRC MSS. However only four samples were studied. These results are not convincing. Immune checkpoint blockade has already been tested in MSS CRC previously treated with oxalip and the results have not been impressive. The authors should comment on this aspect. If they are convinced that oxalip affects the status of MSS CRC thus potentially favoring immune responses the authors should investigate in depth a large number of clinical samples. As CRC is a common disease and oxalip is frequently used in this setting, this analysis should be double.

Minor Points:

- 1) Representative IVIS image of the experiments in figure 1, 3 and 4 should be included (at least in the supplementary figures). The analysis of the graph shows small differences in tumor growth. The images could support the author's hypothesis.
- 2) In the legend of the figure 5a, MIs1/Msh2 should be Mlh1/Msh2.
- 3) In figure 4b the oxali & aPD-L1 arm was interrupted after 35 days. May the authors comment why they euthanized mice only in that specific arm?
- 4) In figure 4b the oxaliplatin arm is missing and should be included to allow proper interpretation of the experimental hypothesis.

Reviewers' comments and our reply:

Firstly, we would express our great appreciation for the reviewers' professional and pertinent comments on our manuscript. These comments and suggestions really helped us a lot in improving the quality of the manuscript.

Both reviewers concerned about assigning MC38 and CT26 cells as MSI and MSS type, respectively. Previously reported whole genomic analyses on these two cell lines showed that there were missense mutations in MMR genes of MC38 cell line, while there were no mutations in MMR genes in CT26 cell line. MSI results when there is MMR-deficiency in a tumor, therefore, MC38 was regarded as MSI type in some reports, while CT26 is regarded as a MSS type. However, since there are no direct results on the microsatellite gene status of these two cell lines, it would be more accurate to state MC38 as a MSI/MMR-deficient cell line, and CT26 as a MSS/MMR-proficient cell line.

In fact, clarification on the whole genome status of the MC38/CT26 cell line is not the emphasis of this study. Previous reports have provided enough evidence to support that CT26 cell line is a reliable model for MSS/MMR-proficient cancer study, and we also found that orthotopic tumor established from this cell line was refractory to PD-L1 inhibitor treatment. Therefore, the idea of this study starts from how to help PD-1/PD-L1 inhibitor therapy to work on this kind of cancer. Unfortunately, our original manuscript did not make this point clear and misled the reviewers to a tangential direction. We apologize for the mistake.

In the revision, we made substantial changes on the manuscript to make the emphasis of this study clearer. By combination of immunogenic chemotherapy and locally expressed PD-L1 trap protein, we solved the two major problems of current PD-1/PD-L1 based immunotherapy: low response rate and immune-related adverse effects. Due to shortage of other similar murine colorectal cancer cell lines, we expanded the test to other known non-hypermutated cell lines (4T1 and B16F10), and proved that the combination was still effective in these tumors. Importantly, in all these models, the locally expressed PD-L1 trap did not induce autoimmune syndromes as did in anti-PD-L1 mAb treatment. We hope the revised manuscript has made the spirit of the intended direction clear.

The following are the point-by-point responses to the comments.

Reviewer #1 (Remarks to the Author):

This work suggests that oxaliplatin may have the potential of changing the tumor microenvironment from an immune excluded tumor to an inflamed tumor, making it susceptible to immune checkpoint inhibition. Several articles have already shown that OXP can induce immunogenic cell death in this model. The authors of this article characterize the immune infiltrate in CT26 tumors following OXP treatment, and explore the combination with immune checkpoint inhibition of PD-L1 to unleash the OXP-induced antitumor immune response. They show that, indeed, the combination appears to have synergistic antitumor efficacy and this is more evident when they combine OXP with the PD-L1 trap, rather than anti-PD-L1.

Major points:

1. Although the article is nicely written and addresses an important question, the previous articles published regarding the immunogenic cell death induced by OXP partially decrease the novelty of the findings. In addition, the authors do not provide any additional mechanistic data.

Reply: Thank you for the pertinent comments on our manuscript. As you mentioned, immunogenic cell death induced by OxP has been reported. It provides a good foundation for our study. In this study, we emphasize on whether OxP induced immunogenic cell death could help to boost the efficiency of PD-1/PD-L1 based cancer immunotherapy, especially in non-hypermutated MSS/MMR-proficient cancers. We found that OxP treatment could enhance antigen-recognition efficiency and turn the “cold” tumor into “hot”, which provided a good condition for applying PD-L1 inhibitor. Since OxP is one of the most widely used first-line chemo drug in cancer chemotherapy, if it could really help PD-1/PD-L1 based immunotherapy, it would be more meaningful than developing a completely new drug. **Thus, the novelty is not the drug itself, but rather the drug’s activity to help immunotherapy.**

2. Another concern is the fact that the authors extrapolate a conclusion for all MSS tumors based on one model, CT26. Although they describe it is poorly immunogenic, this tumor was induced using a carcinogen and, hence, harbors a good number of non-synonymous somatic mutations and potential neoantigens (Castle et al. BMC Genomics 2014, 15:190). Does this model represent well the biology of MSS tumor and its potential immunogenicity, and is one model enough to extrapolate these

findings? It would be interesting to see if other poorly immunogenic tumors (B16) could also be rendered susceptible to immune checkpoint inhibition through combination with OXP. This would make the findings more relevant.

Reply: Really appreciate for these comments and suggestions.

Firstly, as you mentioned, CT26 harbors a good number of non-synonymous somatic mutations and potential neoantigens, however, whole genomic characterization showed that none of the MMR genes (Mlh1, Mlh2, Mlh6, Msh2, Myh, Pms1, Stk1, Mutyh and Ctnb1) were mutated in CT26 cell line (Castle et al. BMC Genomics 2014, 15:190). Since MSS cancers are generally MMR-proficient, and low T-cell infiltration was observed in the established CT26 tumor, we used the CT26 tumor model to represent the MSS colorectal cancer in our original manuscript. To be more accurate, we changed the statement of CT26 tumor model as “MSS/MMR-proficient tumor” in the revised manuscript.

Since there are not many murine colorectal cancer cell lines (CT26 and MC38 are the only two widely used cell lines), we expand our study to other non-hypermutated cancer models (B16F10 and 4T1). Both cell lines are defined as weakly immunogenic cell lines with much less single nucleotide point mutations than others. Tumors established from both cell lines are refractory to PD-L1 inhibitor therapy, while the combination with OxP greatly improved their responses to PD-L1 inhibitor therapy (Fig. 6 in revised manuscript). These results confirmed that the combination with OxP would help the non-hypermutated MSS/MMR-proficient tumors more susceptible to PD-1/PD-L1 immune checkpoint inhibition.

3. Immunogenic cell death had already been shown in this model. The authors should reference the previous literature (line 194 and others).

Reply: Thanks for the suggestion. We added the reference (Tesniere et al. Oncogene, 2010, 29:482) at all necessary places.

4. Fig. S3b, did the authors test against irrelevant peptide to evaluate whether the response is peptide specific and not just non-specific? Are there internal positive and negative controls to ensure the cells are alive and capable of IFN-gamma secretion?

Reply: Thanks for both questions.

The immune response initiated by OxP is peptide specific. When an irrelevant β -galactosidase (β -Gal) peptide was used to pulse the splenocytes of mice after OxP treatment, much fewer IFN- γ spots were observed compared to the AH-1 peptide pulse (**Fig. 2** and **Fig. S2** in revised manuscript).

Before plating splenocytes into the ELISpot plate, we used Trypan blue for live/dead cell staining, and counted only the live cells. IFN-spots are observed in the plates after peptide pulse, therefore, we think these cells are alive and capable of IFN- γ secretion.

5. Additional IHC data on human tumor samples examining the expression of effector cytokines, PD-L1 or other markers would provide stronger evidence that OXP induces immunogenic cell death in human CRC MSS samples.

Reply: Thanks for the suggestion. The human CRC samples are from patients endured 2-3 courses of neoadjuvant chemotherapy of XELOX (OxP included in the regimen), since it has been a period of time after OxP treatment, we don't think ICD markers (such as CRT exposure or HMGB1 release) can still be seen in these samples. In the revised manuscript, CD3 immunofluorescence staining of ten colorectal cancer patient samples was supplemented. In contrast to the samples from untreated patients, profound T-cell infiltration can be seen in the samples from patients endured XELOX treatments (**Fig. 7b**).

6. How many times were each of the *in vivo* experiments repeated? How reproducible is the *in vivo* data?

Reply: All the *in vivo* experiments were repeated at least once, and we draw conclusion only when the results are reproducible.

Minor points:

7. What is the control in Figure S3c? It is not clear in the text nor figures.

Reply: Thanks for the question. The control group is mice without pre-treatment. We labeled it clearly in the revised manuscript (**Fig. S2d**).

8. The statement in line 218: "However, the abundant checkpoint proteins and immunosuppressive

cytokines inside the ELSs definitely impaired the biological functions of T cells inside the OxP treated tumors” is very strong. The data is suggestive, but not conclusive.

Reply: Thanks for the suggestion. We changed this sentence to “the abundant checkpoint proteins and immunosuppressive cytokines may impair the biological functions of T cells inside these ELSs” in the revised manuscript.

9. Fig. 4g and h; It would be interesting to display the values for all the groups, rather than for selected groups, to evaluate the contribution of each of the therapies individually and in combination.

Reply: Thanks for the suggestion. We made major amendments on the organization of the manuscript. Especially, we summarized the PBS, α -PD-L1, OxP and OxP+ α -PD-L1 results in **Fig. 3**, and PBS, PD-L1 trap, OxP and OxP+PD-L1 trap results in **Fig. 5**.

10. The authors make the observation that PD-L1 expression is induced in vivo despite lack of T cell infiltration in CT26. What drives PD-L1 expression if not IFN-gamma? Macrophages? Other cells?

Reply: Thanks for the question. We tested the PD-L1 levels in the established CT26 and MC38 tumors to show that the PD-L1 level is not closely associated with the responsiveness to checkpoint blockade. This was also observed in other studies that higher T cell infiltration, but not PD-L1 level, is more closely associated with responsiveness to checkpoint inhibitors (Tang H. et al. Cancer cell, 2016, 29:285; Ebert et al. Immunity, 2016, 44: 1). Besides tumor cells, PD-L1 are also expressed on dendritic cells and other lymphoid cells. I think that’s the reason for the differences between *in vivo* and *in vitro* PD-L1 expression. Since PD-L1 expression is not the key point in our study, we deleted this part in the revised manuscript.

Reviewer #2 (Remarks to the Author):

In their manuscript, Dr. Song and colleagues investigate the consequences of oxaliplatin-mediated sensitization to immune checkpoint treatment. The authors show that in a colorectal murine cell lines (CT26) the anti-PD-L1 treatment does not affect a host-mediated immune-control. The opposite was obtained when they combine oxaliplatin and PD-L1 modulator owing to an HMGB1 mediated immunogenic cell death. In principle these are interesting findings, however the preclinical models they used has limitations as discussed below. Some of the experiments are not convincing or lack proper controls. Furthermore the implications in the relevant clinical setting (MSS CRC) are not convincing.

1. The authors define MC38 and CT26 as an MSI-H and MSS cell line respectively. They claim that “mismatch repair genes Msh3 as well as Pold1 were mutated in MC38 cell line, indicating this cell line can be used for a valid model for studying MSI-H colorectal cancer”. Although the authors mention a reference to sustain this claim, the paper they reference is unclear and unconvincing (Efremova et al, bioRxiv, 2017). In 2014 Yadav et al published a manuscript in Nature where they genetically characterized MC38. The latest should be used as a reference.
2. To make the claim that MC38 is MSI, a microsatellite analysis should be performed, this should be coupled to NGS. PCR primers to detect microsatellite shift in mouse cells using the matched mouse background are available. The functional impact of alterations in mismatch repair genes should also be validated.

Reply: Really appreciate for your professional and pertinent comments.

I think your major concern in these two questions is the definition of MC38 and CT26 cells as a MSI-H and MSS cell line respectively. Firstly, we apologize for the crude classification of these two cell lines. As you mentioned, previous genomic analysis showed that there are missense mutations in MMR genes in MC38 cell line, and there are no mutations in MMR genes in the CT26 cell line, but since there are no direct microsatellite analysis performed on these two cell lines, it's not accurate to define them as a MSI-H and MSS cell line, respectively. Thank you for pointing out this mistake.

NGS analysis on these cell lines are a bit out of our ability. As described above, this study emphasized on whether OxP treatment could help tumors refractory to PD-L1 inhibitor therapy to become sensitive. These tumors are mostly non-hypermutated MSS/MMR-proficient tumors with low T-cell infiltration. In the revised manuscript, we concentrated on the known non-hypermutated/MMR-proficient tumor types (CT26, B16F10 and 4T1), and tested whether OxP would help PD-L1 inhibitor to work in these tumor models. Along the spirit of these directions, we made major revisions of the manuscript, and hope the revision would make the emphasis more clear than before.

3. The amount of somatic mutations (tumor mutational burden) in MC38 and CT26 could greatly impact immune-control and immune surveillance. The authors do not mention the mutational load of CT26 and MC38 that are two chemically induced tumors. The mutational burden of the actual MC38 and CT26 used in this study should be measured by NGS and inserted in the paper.

Reply: Thanks for these suggestions.

We carried out this study based on the NGS analysis of these cell lines reported previously. The mutational loads are as following:

MC38: 5931 single nucleotide variations (SNVs), 2743 of which were non-synonymous, analyzed by Efremova M. et al. in 2018 (Efremova M. et al. Nature Communication, 2018, 9:32); 4285 non-synonymous variants, among which 1290 are expressed mutations, analyzed by Yadav et al. in 2014 (Yadav et al. 2014, Nature, 515: 572). MC38 showed mutations in the MMR gene Msh3, as well as in Pold1.

CT26, analyzed by Castle et al. in 2014 (Castle et al. BMC Genomics 2014, 15:190): 3023 high-confidence SNVs, 1688 of which cause non-synonymous protein changes. None of the MMR genes Mlh1, Mlh2, Mlh6, Msh2, Msh6, Pms1, Stk1, Mutyh and Ctnnb1 are mutated in CT26.

B16F10: 908 SNVs, analyzed by Castle et al. in 2014 (Castle et al. 2014, Scientific reports, 2014, 4:4743).

4T1: 293 SNVs, analyzed by Castle et al. in 2014 (Castle et al. 2014, Scientific reports, 2014, 4:4743).

There may be some differences between different analyses on the same cell line, but these results generally support the notion that MC38 is a hypermutated/MMR-deficient cell line, while CT26, B16F10 and 4T1 cells are non-hypermuted/MMR-proficient cell lines.

3. The authors show tumor shrinkage of MC38 after PD-L1 administration. The consequences of oxaliplatin treatment alone on MC38 are of interest. Have MC38 tumors complete rejection?

Reply: Thanks for the question. A small dose of OxP (6.0mg/kg) used in this study has modest effect on MC38 tumor growth (**Fig. R1**). It may further help to increase the immune cell infiltration in the tumor, however, since MC38 tumor model is not the emphasis of this study, we did not carry out further study on this model.

Fig. R1 OxP therapy on orthotopic MC38 tumor model (n=4). ns, not significant.

4. An interesting section of the manuscript regards the use of PD-L1 trap plasmid that delivered via liposome-protamine-DNA nanoparticles, locally and transiently, produces PD-L1 blocker in tumor tissue. The in vivo experiments are well conducted but they lack of appropriate controls. Indeed, a pcDNA3.1 backbone should be matched with the PD-L1 trap and in vivo injected. This could help the authors discriminating for an immune response against the plasmid (exogenous DNA vectors and the proteins they express have been shown to be immunogenic) and the nanoparticles present in the system.

Reply: Really appreciate for these comments and suggestions. We supplemented the LPD loaded pcDNA3.1 backbone with GFP sequence as a control of the LPD loaded PD-L1 trap plasmid in the revised manuscript. Neither LPD-pGFP nor LPD-pPD-L1 showed any tumor inhibition effect or induced T-cell infiltration in the tumor (**Fig. 5b**). Therefore, the DNA vector and the nanoparticles should not be the reason for the immune responses.

5. The authors further show that Oxaliplatin induces immunogenic cell death in CT26-FL3 Tumor. Though of putative interest, this working hypothesis is not original since other publications, have already reported oxaliplatin mediated immunogenic cell death. Tesniere and Kroemer in an Oncogene paper (2010), which the authors properly acknowledge, described the HMGB1-dependent activation of the immunogenic cell death in CRC post oxaliplatin treatment. Overall the concept that CRC cell lines undergo immunogenic cell death upon oxaliplatin treatment is not novel.

Reply: Thanks for the comment.

Please see our rebuttal at the beginning of this document. We admit that the concept of CRC and a lot of other cancer cell lines undergo immunogenic cell death upon OxP treatment is not novel, and thanks to Tesniere and Kroemer's excellent work, the mechanism for OxP induced immunogenic cell death has been clarified. However, the emphasis of this study is to test whether OxP-induced immunogenic cell death provides a good condition for applying PD-L1 inhibitor therapy, especially in non-hypermethylated tumors. Since OxP is one of the most important and widely used first-line chemo drugs for cancer therapy, if OxP really works for boosting immunotherapy efficiency in tumors refractory to PD-1/PD-L1 treatment (as shown in this work), that would be much more meaningful than developing a completely new drug. Therefore, the basis of this work is OxP can induce immunogenic cell death in many cancer cell lines, and we further clarified the following points:

1) We showed the value of OxP-induced immunogenic cell death in boosting immunotherapy for MSS/MMR-proficient colorectal tumors as well as other non-hypermethylated tumors.

2) We clarified that the combination has synergistic antitumor effect, since both OxP-induced T-cell infiltration and PD-L1 blockade are necessary for an effective immunotherapy.

3) A locally expressed PD-L1 trap system was applied here, and we proved that the combination of OxP and the PD-L1 trap was an effective, yet safe, method in cancer immunotherapy.

6. Potentially a most interesting (and translational) part of the manuscript is the one dealing with the effect of the oxaliplatin on MSS CRC patients. In this setting the therapeutic choice is limited and immune-checkpoint blockade have failed, most likely due to the low immunogenicity of MSS CRCs. The authors analyze public databases and make the claim that treatment oxalipl (capecitabine plus OxP in fact) affects the status (appearance of ELS structures) of CRC MSS. However only four samples were studied. These results are not convincing. Immune checkpoint blockade has already

been tested in MSS CRC previously treated with oxalip and the results have not been impressive. The authors should comment on this aspect. If they are convinced that oxalip affects the status of MSS CRC thus potentially favoring immune responses the authors should investigate in depth a large number of clinical samples. As CRC is a common disease and oxalip is frequently used in this setting, this analysis should be double.

Reply: Really appreciate these comments and suggestions.

You raised a very good question on why pembrolizumab still failed in MSS CRC patients who have endured at least two courses of previous therapies (We think the paper you mentioned was published by Le et al. in 2015: NEJM, 2015, 372: 26). Indeed, one would expect that some of the patients may have gone through the OxP therapy and should positively respond to the checkpoint inhibitors, but they did not. The major differences between our results and that of the Le et al are as follows:

1) We carried out PD-L1 blockade and OxP combination therapy at the same time, while the patients recruited in pembrolizumab study were with treatment-refractory progressive metastatic cancer. Time course may be an important point in PD-L1 inhibitor and OxP-based chemotherapy. In a pilot study on the immunogenicity of DC vaccination during OxP-based chemotherapy, stage III colon cancer patients receiving standard OxP/capecitabine (XELOX) chemotherapy were vaccinated at the same time with KLH and CEA-peptide pulsed DCs, and enhanced T-cell reactivity upon OxP administration was observed (Lesterhuis et al. British J. Cancer, 2010, 103:1415). This study suggested the immunogenic effect of OxP in colorectal cancer patients. Should PD-L1 inhibitor be used as a first-line treatment together with OxP-based chemotherapy, like that applied in metastatic non-squamous non-small cell lung cancer? In CRC patient sample analysis, profound T-cell accumulation was observed in the tumors from patients endured 2-3 courses of OxP-based chemotherapy, suggesting a good condition for applying PD-1/PD-L1 based immunotherapy. We think this is what should be done, but only further clinical trials can give the answer.

2) Dose is an important factor in combination of chemotherapy with immunotherapy. High dose of chemo drugs as used in clinical regimens are generally harmful to the immune system, and may impair the efficiency of cancer immunotherapy. In our study, we used a relative low dose of OxP in inducing *in vivo* immune response. Of course, what is the optimal clinical dose is still an open question.

3) The tumor mutation burden is an important criterion for successful immunotherapy. Although we proved that OxP can promote antigen-recognition efficiency, which helps to inducing immune responses in relatively non-hypermuted tumors, it is still hard to establish a clinical standard to

determine what extent of mutation burden is enough for OxP to induce an effective immune response. Recently, studies revealed that the neo-antigen ratios are actually the real criterion in inducing effective anti-tumor immune response. In the analysis of MC38 cells by Yadav et al. in 2014, they showed that of the 1290 amino acid changes in MC38 cells, only 7 were presentable by MHCI (Yadav et al. 2014, Nature, 515: 572).

We hope these considerations will help to resolve the apparent discrepancy between our study and that of Le et al, as well as the further clinical trials of combination of OxP-based chemotherapy and PD-1/PD-L1 based immunotherapy. We have added a discussion on these points in the discussion part of the revised manuscript.

Additionally, we assessed ten MSS CRC patient samples and performed CD3 immunofluorescence staining of these samples. The data have been included in the revised manuscript. In the samples from patients endured XELOX treatment, profound T-cell accumulation was observed (**Fig. 7b**). The results are consistent with those reported in the original manuscript using anti-CD3 immunohistochemistry with samples obtained from only 4 patients. Again, the data suggest that OxP-based chemotherapy may provide an excellent opportunity for the checkpoint inhibitors in MSS CRC patients.

Minor Points:

1) Representative IVIS image of the experiments in figure 1, 3 and 4 should be included (at least in the supplementary figures). The analysis of the graph shows small differences in tumor growth. The images could support the author's hypothesis.

Reply: Thanks for the suggestion. We supplemented the IVIS images and tumor images for all the treatments in the supplementary materials (**Fig. S4, Fig. S7, Fig. S13**).

2) In the legend of the figure 5a, Mls1/Msh2 should be Mlh1/Msh2.

Reply: Thanks for the reminder.

3) In figure 4b the oxali & aPD-L1 arm was interrupted after 35 days. May the authors comment why they euthanized mice only in that specific arm?

Reply: Thanks for the question. We stopped most of our observation on day 35, and only carried out prolonged observation for the Oxp+PD-L1 trap group. To make this point clear, we uniformed all the tumor growth curves to stop at day 35.

4) In figure 4b the oxaliplatin arm is missing and should be included to allow proper interpretation of the experimental hypothesis.

Reply: Thanks for the suggestion. We made major amendments on the organization of the manuscript. Especially, we summarized the PBS, α -PD-L1, Oxp and Oxp+ α -PD-L1 results in **Fig. 3**, and PBS, PD-L1 trap, Oxp and Oxp+PD-L1 trap results in **Fig. 5**.

Reviewers' comments:

Reviewer #1 (Remarks to the Author):

The authors have addressed all of my questions and have rewritten the article to include those modifications. In addition, the current version better highlights the importance of the findings in multiple tumor types and novelty of the findings, which was my main concern.

Reviewer #2 (Remarks to the Author):

In this revised manuscript, the authors have addressed a number of concerns raised during my initial review. The newly added experiments, such as the novel cell lines that were included, are a useful addition and address some of the technical concerns. On the other hand, one main issue does remain. In the rebuttal the authors declare: "we assessed ten MSS CRC patient samples and performed CD3 immunofluorescence staining of these samples. The data have been included in the revised manuscript". This analysis is not clear in the related chapter and legend, I suggest to describe clearly that the authors analyzed ten patients in the manuscript and to show a plot for T cell infiltration in all patients. This is relevant for the impact of the manuscript. In addition I remain concerned by the lack of in depth characterization of the MMR genes status of the cell lines. Although this is not the main focus of the paper, it would add significantly to the potential clinical impact of the story.

Point-by-point response to the referees' comments:

Reviewer #1 (Remarks to the Author):

The authors have addressed all of my questions and have rewritten the article to include those modifications. In addition, the current version better highlights the importance of the findings in multiple tumor types and novelty of the findings, which was my main concern.

Reply: We are glad that the revision answered your concerns. Really appreciate for your help in improving the quality of the manuscript.

Reviewer #2 (Remarks to the Author):

In this revised manuscript, the authors have addressed a number of concerns raised during my initial review. The newly added experiments, such as the novel cell lines that were included, are a useful addition and address some of the technical concerns. On the other hand, one main issue does remain. In the rebuttal the authors declare: "we assessed ten MSS CRC patient samples and performed CD3 immunofluorescence staining of these samples. The data have been included in the revised manuscript". This analysis is not clear in the related chapter and legend, I suggest to describe clearly that the authors analyzed ten patients in the manuscript and to show a plot for T cell infiltration in all patients. This is relevant for the impact of the manuscript. In addition I remain concerned by the lack of in depth characterization of the MMR genes status of the cell lines. Although this is not the main focus of the paper, it would add significantly to the potential clinical impact of the story.

Reply: Thank you very much for your continuous efforts in helping us to improve the quality of the manuscript, which is in line with the spirit of reviewing processes for publications in a high impact journal.

1) We apologize for the unclear statements on the analysis of the CRC patient samples. In the 2nd revision, we gave a clear description on the analysis details of the samples in the related chapter (Page 11, "Colorectal cancer patient sample analysis"). We also amended the legend of Figure 7, and provided a separate description for each image or statistic plot.

2) We appreciate for your recognition of the potential clinical impact of our work. As stated in the 1st revision, although we did not carry out whole genome sequencing analysis of the cell lines by

ourselves, these cell lines have been well-characterized in previous publications (*Castle et al. BMC Genomics 2014, 15:190; Castle et al. Scientific reports, 2014, 4:4743; Efremova M. et al. Nature Communication, 2018, 9:32*). All the genomic data are available from the public database: NGS fastq files of CT26, 4T1, B16, BALB/cJ and C57BL/6 are available from the European Nucleotide Archive as PRJEB5791, PRJEB5299, PRJEB5797, PRJEB5321 and PRJEB5312, submitted by John C. Castle et al. in 2014; SNP array data of MC38 were deposited in the GEO under the accession number GSE93018, and the exome sequencing bam files of MC38 were deposited in the Sequence Read Archive under the accession number SRP095725.

Details on the single nucleotide variants (SNVs) and MMR gene status of these cell lines are as following:

CT26: 3023 SNVs. *Kras* is mutated at p.G12D. *Apc*, *Tp53*, *Braf*, *Pold1* and MMR genes *Mlh1*, *Msh2*, *Msh6* and *Pms2* are not mutated.

B16: 908 SNVs. *Tp53*, *Braf*, *Pold1* and MMR genes *Mlh1*, *Msh2*, *Msh6* and *Pms2* are not mutated.

4T1: 293 SNVs. *Tp53*, *Braf*, *Pold1* and MMR genes *Mlh1*, *Msh2*, *Msh6* and *Pms2* are not mutated.

MC38: 5931 SNVs. *Tp53*, *Braf*, *Pold1* and MMR gene *Msh3* are mutated.

The above results indicate that CT26, B16 and 4T1 cells are all non-hypermuted MMR-proficient cells, while MC38 is a hypermutated cell line.

In a recently published paper (*Germano G. et al, Nature, Dec. 2017*), the authors also viewed CT26 as a MMR-proficient cancer cell line. They stated “*To define the functional role of MMR in tumor formation and response to immunotherapy, we studied MMR-proficient mouse colorectal cancer (CT26) ...cells.*”

We believe these data are enough to support our study in using CT26, B16 and 4T1 cells to establish non-hypermuted MMR-proficient tumor models.

To make this point more clear, in the 2nd revision, we provided clear descriptions on the MMR gene status of each cell line, followed by corresponding references (Page 4 for CT26 cell; Page 5 for MC38 cell; Page 10 for B16 and 4T1 cells).